# Reconsidering Spatial Alignment for Longitudinal Breast Cancer Risk Prediction

## Abstract

Regular mammography screening is key for early breast cancer detection, and deep learning enables personalized screening strategies. However, misalignment across time points can obscure subtle tissue changes and degrade risk prediction performance. This study provides insights into the impact of different alignment strategies, namely image-based registration, feature-level alignment, and implicit methods, on risk prediction using two large-scale mammography datasets, offering guidance for future research and methodological development. Results show that our newly proposed image-based registration model outperforms others, improving accuracy and yielding anatomically plausible deformations, underscoring the importance of precise alignment in longitudinal risk modeling.

## 1 Introduction

Mammography remains the gold standard for breast cancer screening [1], and widespread screening has been shown to reduce mortality [17]. However, challenges persist, particularly for individuals at high risk [13]. Recent deep learning studies suggest that incorporating longitudinal mammography, using imaging from multiple timepoints, can enhance risk prediction beyond models based on single-timepoint data [2, 7, 9, 15, 14]. To fully leverage these benefits, accurate alignment of images across time is essential, a task complicated by variations in breast tissue and differences in patient positioning [4]. Alignment strategies are typically categorized as either explicit, where images or features are directly registered, or implicit, where alignment is learned jointly during feature extraction. We perform the first systematic study of alignment strategies for longitudinal breast cancer risk prediction, providing insights into both explicit and implicit approaches. Building on these insights, we propose a new image-based alignment model that achieves improved predictive performance. **Our main contributions are:**

- A unified framework for evaluating explicit (image-/feature-level) and implicit alignment strategies for longitudinal breast cancer risk prediction.
- A novel risk prediction model that leverages image-based alignment to generate anatomically meaningful deformations, achieving state-of-the-art performance on two large-scale datasets.

## 2 Methods

We address the challenge of five-year breast cancer risk prediction by evaluating six temporal alignment strategies within a unified framework (Figure 1).

**No Alignment:** Our baseline builds on prior work [16, 15], combining Multilevel Joint Learning [15], Temporal Self-Attention [10], and a Cumulative Probability Layer [16, 12, 9]. Current and prior images are encoded with a shared backbone, processed via temporal self-attention, and used for risk prediction. Additional prediction heads estimate risk from each timepoint independently (Figure 1a).

**Implicit Alignment:** In this strategy, current and prior images are encoded, and their feature maps

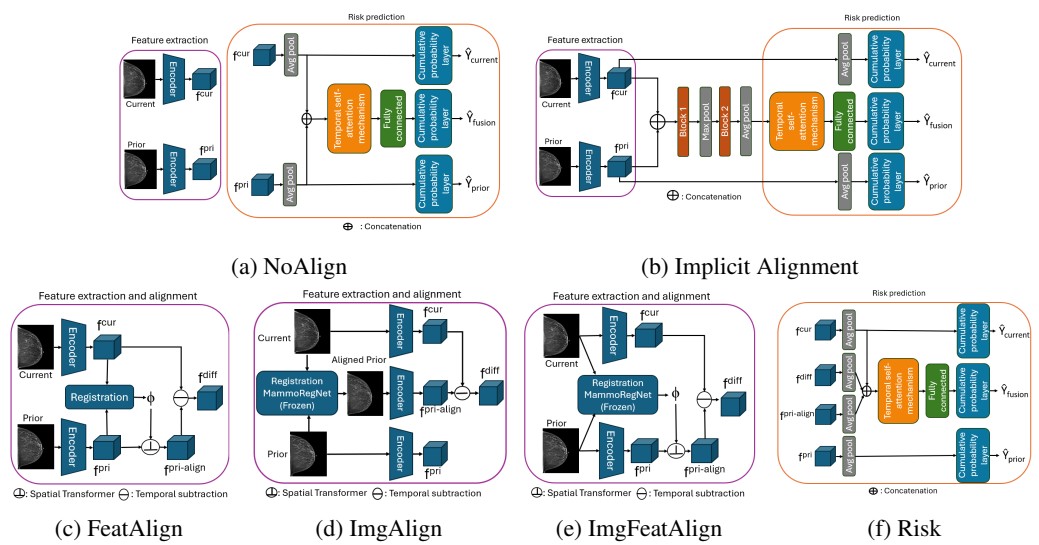

(a) NoAlign

(b) Implicit Alignment

(c) FeatAlign

(d) ImgAlign

(e) ImgFeatAlign

(f) Risk

Figure 1: Overview of longitudinal risk prediction methods: (a) Direct feature extraction without alignment, (b) Implicit Alignment, (c) Feature-level alignment, (d) Image-level alignment with MammoRegNet, (e) Applying MammoRegNet's deformation field in feature space, and (f) Risk prediction using alignment methods (c), (d), and (e).

are concatenated before being processed through convolutional and attention layers. Temporal dependencies are learned implicitly, without explicit spatial alignment (Figure 1b).

**Explicit Alignment:** This approach enhances the risk prediction baseline by incorporating spatial alignment via deformation fields, enabling better temporal feature fusion. It leverages four key feature maps, as in [15]: current $\mathbf{f}^{\text{cur}} \in \mathbb{R}^{C \times h \times w}$, prior $\mathbf{f}^{\text{pri}}$, aligned prior $\mathbf{f}^{\text{pri-aligned}} \in \mathbb{R}^{C \times h \times w}$, and temporal difference features $\mathbf{f}^{\text{diff}} = \mathbf{f}^{\text{cur}} - \mathbf{f}^{\text{pri-aligned}} \in \mathbb{R}^{C \times h \times w}$, which capture temporal changes. Predictions are generated from three representations: Current, Prior, and Fused, where the fused input is formed by concatenating $\mathbf{f}^{\text{cur}}$, $\mathbf{f}^{\text{diff}}$, and $\mathbf{f}^{\text{pri-aligned}}$. The overall risk prediction architecture is illustrated in Figure 1f. We investigate alignment strategies at both the image and feature levels:

**Feature-Level Alignment (FeatAlign / FeatAlignReg):** This method learns a deformation field to align prior feature maps, $\mathbf{f}^{\text{pri}}$, to current feature maps, $\mathbf{f}^{\text{cur}}$. FeatAlignReg introduces smoothness regularization to ensure anatomically plausible deformation fields (Figure 1c).

**Image-Level Alignment (ImgAlign):** As an alternative to feature-level alignment, we propose MammoRegNet, a deep learning-based registration network inspired by the Non-Iterative Coarse-to-Fine Transformer (NICE-Trans) architecture [11]. MammoRegNet is used to align prior mammograms, $\mathbf{I}^{\text{pri}} \in \mathbb{R}^{H \times W}$, to the current ones, $\mathbf{I}^{\text{cur}} \in \mathbb{R}^{H \times W}$. In this setup (Figure 1d), current, prior, and aligned prior images are encoded to extract features, from which temporal difference features $\mathbf{f}^{\text{diff}}$ are computed. These features are then passed to the risk prediction module (see Figure 1f).

**Image-Based Feature Alignment (ImgFeatAlign):** Rather than applying MammoRegNet's deformation field at the image level, this variant applies it directly in feature space (Figure 1e). This setup allows us to explore whether image-driven deformation fields can still improve temporal feature fusion when used post-encoding, potentially benefiting from both anatomically grounded registration and deeper feature representations.

## 3 Experimental Setup

**Datasets:** We evaluate on two large, publicly available mammography datasets. EMBED[1] [6] and CSAW-CC[2] [3]. Following [15], we include patients with $\geq 5$ years of follow-up. Images are resized to $1664 \times 2048$ while preserving aspect ratio and split into training, validation, and test sets (5:2:3).

**Evaluation metrics:** Alignment quality is quantified by the percentage of Negative Jacobian Determinants (NJD) [5], while risk prediction performance is assessed via C-index and AUC for 1–5 year horizons [9, 15, 16], with 95% confidence intervals from 1,000 bootstraps.

[1] https://aws.amazon.com/marketplace/pp/prodview-unw4li5rkivs2#overview
[2] https://snd.se/en/catalogue/dataset/2021-204-1

Table 1: 1–5 year breast cancer risk prediction using different alignment methods. C-index and selected AUC values (1, 3, 5 years) with 95% confidence intervals for both datasets.

| Method | EMBED | | | | CSAW-CC | | | |
|---|---|---|---|---|---|---|---|---|
| | C-index ↑ | 1-yr ↑ | 3-yr ↑ | 5-yr ↑ | C-index ↑ | 1-yr ↑ | 3-yr ↑ | 5-yr ↑ |
| NoAlign | 64.0 (61.7–66.7) | 64.9 (62.1–67.9) | 63.7 (61.2–66.3) | 55.7 (51.4–60.0) | 65.9 (64.0–67.8) | 66.1 (63.8–68.3) | 65.7 (63.8–67.6) | 66.8 (64.5–68.9) |
| Implicit | 70.9 (68.6–73.3) | 72.5 (69.3–75.5) | 69.3 (66.6–71.8) | 65.7 (62.0–69.7) | 67.6 (65.8–69.7) | 68.2 (65.7–70.6) | 68.3 (66.3–70.2) | 68.7 (66.3–71.1) |
| FeatAlign | 72.2 (69.5–75.5) | 72.4 (69.5–75.6) | 72.0 (69.7–74.6) | 68.5 (64.8–72.0) | 69.1 (67.0–71.1) | 70.1 (67.9–72.4) | 70.0 (68.1–71.9) | 71.6 (69.4–73.8) |
| FeatAlignReg | 70.6 (67.8–73.2) | 71.2 (68.3–74.3) | 70.7 (68.2–73.5) | 65.7 (61.7–69.6) | 68.4 (66.4–70.4) | 68.9 (66.7–71.2) | 69.8 (68.0–71.6) | 72.0 (69.9–74.2) |
| ImgAlign | 72.3 (69.6–74.8) | 73.6 (70.6–76.5) | 72.3 (69.8–74.5) | 69.7 (66.2–73.4) | 70.2 (68.1–72.1) | 71.2 (68.9–73.4) | 71.7 (69.9–73.4) | 73.9 (71.7–76.0) |
| **ImgFeatAlign** | **74.7 (72.3–77.0)** | **75.0 (72.1–77.7)** | **75.3 (73.1–77.4)** | **72.5 (68.9–75.7)** | **70.4 (68.2–72.3)** | **72.0 (69.6–74.2)** | **72.6 (70.8–74.5)** | **75.2 (73.1–77.5)** |

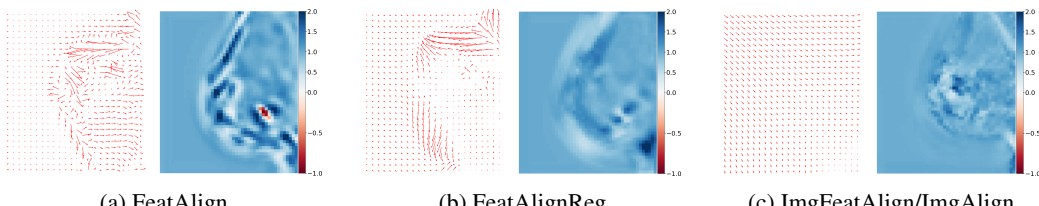

(a) FeatAlign      (b) FeatAlignReg      (c) ImgFeatAlign/ImgAlign

Figure 2: Comparison of deformation field quality. Each method shows displacement vectors (left), and Jacobian determinant maps (right) (white/blue: valid; orange/red: invalid non-invertible regions).

**Implementation Details:** We use the pre-trained Mirai encoder [16], as a frozen backbone. For feature-level alignment, risk prediction and alignment are jointly optimized using L2 feature-matching and binary cross-entropy losses. For image-level alignment, MammoRegNet is frozen, and only the prediction loss is optimized. Models are trained with Adam [8] (LR $1 \times 10^{-5}$, weight decay $1 \times 10^{-6}$, batch size 20) for 40 epochs. Learning rate is halved after 5 stagnant epochs and training stops after 15. Augmentations include affine transforms, color jitter, gamma adjustment, and cropping.

## 4 Results

Table 1 summarizes 1- to 5-year breast cancer risk prediction performance (C-index and AUC with 95% CI) for each alignment strategy. ImgFeatAlign consistently achieves the highest C-index and stable AUC, demonstrating superior predictive strength and robustness over time. FeatAlign performs reasonably well but is consistently outperformed by image-level alignment. The Implicit method shows moderate results, while NoAlign yields the lowest scores, with the steepest AUC decline, underscoring the importance of alignment in longitudinal models. These findings highlight the value of advanced alignment strategies for improving the accuracy and reliability of breast cancer risk prediction.

Figure 2 shows displacement vectors and Jacobian determinant maps for the three registration methods. FeatAlign yields noisy, irregular deformations with invalid (negative Jacobian) regions. FeatAlignReg improves smoothness and invertibility but remains locally constrained. In contrast, ImgAlign and ImgFeatAlign produce smooth, coherent, and anatomically plausible fields with consistent displacements and no invalid regions, indicating higher alignment quality.

## 5 Conclusion and Outlook

In summary, accurate spatial alignment is crucial for longitudinal breast cancer risk prediction. Image-based approaches, especially ImgFeatAlign, achieve superior performance by balancing anatomical precision with high-level feature representation. These findings highlight the potential of robust longitudinal modeling to enhance personalized screening and early intervention. Future work will extend these alignment strategies to broader longitudinal imaging tasks and integrate multimodal data to improve interpretability and risk stratification.

## Potential Negative Societal Impacts

Our method aims to improve personalized breast cancer screening by leveraging longitudinal imaging data to better predict individual risk. This approach has the potential to support earlier detection, reduce unnecessary procedures, and improve patient outcomes. While we do not anticipate any direct negative societal impacts specific to our method, we acknowledge the broader dual-use risks associated with machine learning in healthcare. In particular, the rich information derived from longitudinal imaging data could, in theory, be misused—such as for unauthorized profiling or predicting unrelated health conditions without patient consent.

Additionally, if such models are deployed prematurely or without adequate clinical oversight, they may underperform in underrepresented populations or lead to over-reliance on automated risk scores. This could result in misdiagnoses, over-screening, or under-screening. To mitigate these risks, it is critical to ensure fairness, transparency, and responsible integration into clinical practice.

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
