# OpenReview forum: "Reconsidering Spatial Alignment for Longitudinal Breast Cancer Risk Prediction"
_EurIPS.cc/2025/Workshop/MedEurIPS — EurIPS 2025 Workshop MedEurIPS Submission_

### Official Review · Reviewer_qWn7 · 2025-10-24
**Interesting work on alignment strategies and their effect on downstream risk prediction**

**Rating:** 4
**Confidence:** 4

**Review:**

This paper presents a systematic comparison of spatial alignment strategies for longitudinal mammography-based risk prediction. The authors evaluate both explicit and implicit alignment approaches, and introduce an image-based registration model (MammoRegNet) that achieves anatomically coherent deformations and improved predictive performance. The topic is well motivated, as longitudinal alignment remains an underexplored but critical aspect of temporal modeling in breast imaging, and medical imaging in general.

That said, the methodological novelty is moderate. The study could benefit from deeper discussion of generalizability and dependence on the number of input scans.

---

### Official Review · Reviewer_x83g · 2025-10-31
**This paper, "Reconsidering Spatial Alignment for Longitudinal Breast Cancer Risk Prediction," addresses a critical methodological challenge in deep learning for medical time-series analysis: how to effectively align misregistered images across different patient visits.**

**Rating:** 7
**Confidence:** 4

**Review:**

This study provides a valuable and systematic comparison of alignment strategies for longitudinal mammography-based breast cancer risk prediction. This fills a significant gap in the literature regarding optimal temporal registration methods.

``Strengths``
1. Systematic Evaluation: The paper's primary strength is the comprehensive, unified framework used to evaluate six distinct alignment approaches on two large-scale datasets (EMBED and CSAW-CC).
2. The work effectively demonstrates a crucial finding for the deep learning community: methods achieving high performance (like the proposed ImgFeatAlign) are those that prioritize anatomically plausible deformation fields.

``Minor Suggestions/Areas for Future Work``
The reliance on a frozen, pre-trained backbone (Mirai encoder) limits the exploration of truly end-to-end learning where the image encoding and alignment are optimized jointly.

---

### Decision · Program_Chairs · 2025-10-31

**Decision:**

Accept (Poster)

**Comment:**

Both reviewers find the paper well-motivated and relevant, offering a systematic evaluation of spatial alignment strategies for longitudinal breast cancer risk prediction.